# A WoT Platform for Supporting Full-Cycle IoT Solutions from Edge to Cloud Infrastructures: A Practical Case

**DOI:** 10.3390/s20133770

**Published:** 2020-07-05

**Authors:** Rafael Pastor-Vargas, Llanos Tobarra, Antonio Robles-Gómez, Sergio Martin, Roberto Hernández, Jesús Cano

**Affiliations:** 1Department of Control and Communication Systems, Computer Science Engineering Faculty, Spanish National University for Distance Education (UNED), 28040 Madrid, Spain; llanos@scc.uned.es (L.T.); arobles@scc.uned.es (A.R.-G.); roberto@scc.uned.es (R.H.); jesus.cano@computer.org (J.C.); 2Electrical and Computer Department, Industrial Engineering School, Spanish National University for Distance Education (UNED), 28040 Madrid, Spain; smartin@ieec.uned.es; 3Faculty of Law, San Pablo CEU University, 28003 Madrid, Spain

**Keywords:** web of things, IoT learning, cloud computing, protocols, virtualization, instructional design

## Abstract

Internet of Things (IoT) learning involves the acquisition of transversal skills ranging from the development based on IoT devices and sensors (edge computing) to the connection of the devices themselves to management environments that allow the storage and processing (cloud computing) of data generated by sensors. The usual development cycle for IoT applications consists of the following three stages: stage 1 corresponds to the description of the devices and basic interaction with sensors. In stage 2, data acquired by the devices/sensors are employed by communication models from the origin edge to the management middleware in the cloud. Finally, stage 3 focuses on processing and presentation models. These models present the most relevant indicators for IoT devices and sensors. Students must acquire all the necessary skills and abilities to understand and develop these types of applications, so lecturers need an infrastructure to enable the learning of development of full IoT applications. A Web of Things (WoT) platform named Labs of Things at UNED (LoT@UNED) has been used for this goal. This paper shows the fundamentals and features of this infrastructure, and how the different phases of the full development cycle of solutions in IoT environments are implemented using LoT@UNED. The proposed system has been tested in several computer science subjects. Students can perform remote experimentation with a collaborative WoT learning environment in the cloud, including the possibility to analyze the generated data by IoT sensors.

## 1. Introduction

Internet of Things (IoT) [1] has become a key technology for the interconnection of smart devices [2] with their surroundings. These devices acquire information from their immediate environment using specific sensors and change the state of their environment through actuators. These changes are performed through algorithms that determine the interaction with the environment. This computational capacity is defined by the “Edge Computing” paradigm, which encompasses not only algorithmic solutions but also the boundary conditions that must be taken into account when implementing the device’s intelligence [3,4,5]. These conditions include requirements in terms of response time, cost and energy consumption and use of bandwidth in communications, among others.

In the field of education, these technologies have been employed in computer science courses [6], by allowing students to have a smooth and natural approach to them and their applications [7,8]. Additionally, [9,10] present the evolution of IoT learning scenarios in contexts like distributed computing and cybersecurity. These contexts use distance learning/teaching methodologies and corresponding environments.

The use of IoT applications has multiple fields of application [1], such as e-Health (health monitoring of people [11], Personalized Healthcare [12] or biosensors-based environments [13,14]), Smart Cities (traffic control [15] or intelligent transport systems [16]), Agriculture [17,18] or the vehicle industry [19,20], among many others. The applications are practically endless, considering that the number of intelligent devices and sensing systems are growing at a dizzying pace.

IoT has exploded in recent years, and it does not look like a short-term slowdown is taking place. Gartner [21] predicted that there will be 20.4 billion smart devices connected and in use worldwide by 2020, and a new Business Insider Intelligence study [22] predicts that the IoT market will grow by more than $3 billion a year by 2026.

Taking into account the need for professionals in all the areas mentioned above, it is necessary to have specific learning processes that allow students to acquire the necessary competences and skills to undertake projects based on IoT infrastructures. Students must use components and layers (hardware/software) that are deployed in this type of solution, so the learning process must incorporate the use of technological tools similar to those that will be found on these IoT environments and domains. Thus, the objectives of this paper are the following:Analyze the main stages involved in the IoT development cycle and define the essential characteristics of an environment that supports the learning and experimentation of all these stages.Describe the main features of a system designed by the authors that cover all IoT development stages and and how this system fulfils the essential characteristics mentioned before.Evaluate the students’ perception of the platform’s usefulness and its applicability in the different stages involved in IoT projects.

The developed platform, Labs of Things at UNED (LoT@UNED), provides remote laboratories for full IoT development, including edge, fog and cloud computing and complemented with communication protocols and cybersecurity. The use of these remote laboratories allows students to acquire complete IoT skills using real devices and platforms from home. The paper also introduces its use in an official master degree in Computer Engineering.

Regarding the paper organization, Section 2 shows the methodology followed in this paper. Section 3 describes the state of the art found in the literature about IoT remote laboratories. Section 4 describes the platform proposed by the authors, from the hardware, software and communications point of view. Section 5 describes the practices implemented with this platform in a real use case. Section 6 provides the results of a satisfaction survey provided to students. Section 7 details the discussion of the main findings from the survey. Finally, conclusions are given in Section 8.

## 2. Methods

The methodology followed in this study includes the following steps:Analysis of IoT applications to determine the main stages of IoT development.Analysis of the literature to identify previous papers published describing IoT remote laboratories. This analysis consists of the search for articles in the main scientific repositories for this topic: MDPI, IEEExplorer and ScienceDirect. This step analyzes in which stages of IoT development are focused the found papers.Analysis of the system proposed by the authors to check the stages of IoT development covered.In-depth description of the proposed system by the authors from the hardware, communications and software points of view.Description of the experimentation of this platform in a real Computer Engineering subject, including the full IoT development cycle and indicating the designed practices provided to the students. A methodology based on a typical flow of the instructional design [23] has been used. The first assumption made to start the application of instructional design is that the setting up of a laboratory is not an isolated task. It should be integrated into the subject objectives as another element of the instructional design. It should be a way to acquire a competence or skill related to the subject. In addition to this, the methodology involved in a teaching/learning process on distance, as our case is, implies much more periodical virtual attendance and interaction with/among students than with a traditional methodology. All these learning resources presented to students have to be available to them all time, and they expect innovation teaching approaches to improve the quality of courses. This expectation is more noticeable in Engineering subjects than only theoretical subjects such as maths when practical skills have to be acquired by students. Additionally, supporting many students becomes a real challenge when technologies are implemented and deployed in virtual courses. The instructional design methodology is made up of the four phases, as it can be observed in the Figure 1:
*Activity Description*. The educational objectives are first defined, a global description of the laboratory is given to students and the expected outcomes are detailed to them in this step.*Activity Design*. In this phase, a set of elements to be employed in the activity are selected, the acquisition data mechanism and the way in which the elements interact.*Activity Development*. Once the laboratory design is finished, students will be required to perform some programming task with a set of provisioned services and obtained data, as well as running a set of client applications. These have to be used, tested and synchronized among them.*Experimentation*. The last step is to do experimentation with the services and applications deployed to make improvements.Survey preparation to analyze students’ satisfaction. It includes questions about: gender, age and occupation. It also includes five-point Liker-scale questions about perceived usefulness, ease of use, user attitude, social influence, ease of access and intention of use.Analysis of the satisfaction survey provided to the students to validate the tool from a satisfaction point of view. The previously mentioned indicators are analyzed by studying their standardized mean, standard deviation, variance, minimum and maximum values, median, kurtosis, asymmetry and Cronbach’s alpha.

## 3. State of the Art

To understand the complexity associated with the development of IoT solutions, it is important to understand the organization of these systems, usually in a set of layers that implement specific functionalities [3,24], as it can be observed in Figure 2. Usually, these layers are classified using a criterion of physical proximity to the environment and processing capacity of the components that integrate it [25]:Layer 1: Edge computing [26,27]. This layer integrates hardware and software components as smart devices, sensors and IoT protocols.Layer 2: Fog computing [4,5]. This intermediate layer provides an extra resources layer, such as computing power or real-time services, to the edge layer.Layer 3: Cloud layer or dashboards [28] and assisted decision layer [24,29]. Data based solutions using cloud services and the storage service of sensor data in layer 2. Usually, this layer uses services, which need a high computational power capacity so this layer can be integrated with the cloud provider of layer 2 or be located in another cloud provider, such as AWS IoT [30], Microsoft Azure IoT [31] and IBM Watson IoT [32], or specific platforms [33].

Other important aspects to have into account when analyzing IoT applications are the communication technologies and protocols (such as HTTP, MQTT [25], CoAP [35] and others [36,37,38]), and cybersecurity.

Analyzing the literature, most of the educational IoT labs are for hands-on experimentation. Among those designed to experiment online out of laboratory facilities, many were just pure simulations or virtual labs [39]. An example is the work of Patil et al. [40], who describe an IoT virtual lab to allow students sensing and retrieving simulated data from the cloud using Python, as part of the modeling and simulation lab.

Only a few remote labs can be found to allow remote experimentation. An example is the work of Tunc et al. [41], who presented an IoT remote laboratory designed only for cybersecurity experimentation.

El-Hasan [42] introduces an IoT mobile dashboard to allow off-campus practices through a system including sensors, controlling and interfacing kits, cameras and others. Basically it only allows the modification of certain parameters to switch the direction of rotation of a motor by changing predefined values of voltage, current and power as well as other required parameters, such as speed and torque.

Fernandez-Pacheco et al. [43] describe an Arduino remote lab using a Raspberry Pi as a server, but it is only intended for microcontroller programming (Arduino), not for IoT purposes (cloud, Python programming, IoT protocols, cybersecurity, etc.).

Leisenberg [44] presents a remote lab based on Raspberry Pi for movement analysis. Students should write the code to analyze real time images coming from a webcam. Again this system is not intended for full-cycle IoT purposes.

Rajurikar et al. [45] present a system for IoT protocols experimentation, mainly REST and CoAP. They connected several sensors through an Arduino board and a Beagle Bone device to a cloud platform. This cloud environment is only intended for End point Data Acquisition and decision-making.

From the previous work, we can conclude the essential characteristics to be covered in a IoT laboratory/environment are:Access and development with IoT devices (edge programming).Development of solutions in the fog, with limited computational capacities (fog programming).Analysis of the data provided by the sensors and interaction with the actuators of the device infrastructure (cloud dashboard and analytics programming).Configuration and management of specific communication protocols for IoT (protocol experimentation).Development of security techniques in IoT environments (cybersecurity).

It can also be observed that none of the analyzed works allow the implementation of all the essential features required for learning the complete development cycle of the IoT applications. Only Rajurikar et al. [45] complete the implementation and support of three out of the five features. As a consequence, it is necessary to have an environment that complies with all the characteristics.

Table 1 compares the functionality implemented in each of the IoT remote labs found on the literature and the authors’ proposed system. It can be observed that our proposal covers all the features included in the study (edge programming, fog programming, cloud dashboards and analytics programming, protocol experimentation and cybersecurity), whereas other approaches only cover one or several functionalities. This way, the development of LoT@UNED implements the full set of features, advancing in the development/research of this type of environments.

To check how these features are implemented in the authors’ proposed system, the following section describes the LoT@UNED platform more in detail.

## 4. Solution Description

### 4.1. Hardware Architecture

The LoT@UNED platform implements the edge layer through a set of IoT devices (i.e., Raspberry Pi boards). Each device is connected to the services of layer 2 (Cloud IoT Layer) using the MQTT protocol. This way, students can develop the skills and abilities corresponding to layers 1 and 2. The services of the Cloud IoT layer are provided through the IBM cloud provider, and specifically using the IBM Watson IoT service. The service for storing sensor/device data is also implemented in this provider. The non-relational database Cloudant is used for this purpose [46]. This cloud storage service is used in the dashboard and assisted decision layer (Cloud Layer). Again, the IBM Watson Studio service from IBM Cloud is used for the development of the analysis and machine learning algorithms based on the data stored in the Cloudant service. LoT@UNED has been designed flexibly to be able to use other specific services from other providers for the cloud layers, so that experiences can be designed for the development of IoT solutions using the AWS or Google services (AWS IoT, Cloud IoT, S3 or Cloud Storage). The basis of a previous generation of our Web of Things (WoT) platform was presented in [10].

The structure of the platform focuses on the availability of low-cost Raspberry Pi devices, which are connected through a cluster that eases the device connection to the Internet. There are two logical groupings available that facilitate the connection of new devices. The first group uses a specific rack for the management and connection of the devices, as shown in Figure 3a. The characteristics of the rack can be found on the website of the BitScope provider [47]. It allows grouping up to 40 Raspberry Pi (Model B) devices, facilitating the connection to the electricity grid, the connection among devices and the Internet connection. It can also be installed in a traditional rack, facilitating the management of the cluster itself. Its high cost and the inability to add specific sensors in an individual way for each device can be noticed as its main drawbacks.

The second grouping, as shown in Figure 3b, uses cheaper and more flexible components in terms of device separation. This fact allows us to add specific sensors (cameras, GPS, temperature sensors, etc.) without storage problems or cluster connectivity. In fact, the storage can be increased by lateral fixings that support the structure of the cluster. Proof of this is the specific configuration that is used in the example specified in the following section. This type of configuration is deployed outside of the clusters to ease the replacement and management of the sensors in order to provide the necessary redundancy for the services that uses this configuration.

The variety of setups (clustered or individual) allows the logical grouping of the services offered by LoT@UNED and, also, the redundancy necessary to provide a stable learning service for students. For example, in the case of the setup of Figure 5, there are three exact replicas that are managed by the software developed, installed on the base image of the Raspberry Pi and integrated transparently within the service availability (a service, three concurrent accesses). The base image of each Raspberry Pi card comes with the connection services to the IoT service in the cloud, which allows the self-registration of the devices. This self-registration allows us to automatically have the inventory of the devices and the available setups for the entire IoT environment of LoT@UNED. Each device will add specific information about the type of educative service offered (it can be more than one), which will allow the activity manager to decide on the assignment of each environment for the student who requests it.

### 4.2. Communication

Resources in LoT@UNED are understood as a standard communication channel using the MQTT protocol (for interaction commands) and the required software to “control” and “program” the device (a Python distribution, sensor access libraries, etc.). All these resources define a run-time environment that depends on the service that the end-user wants to offer.

The MQTT protocol has been selected due it its popularity and specific way of working. This paradigm uses the message as a fundamental unit of communication. The participants in the solution are the ones who give meaning to the message. The roles of subscriber, editor (publisher) and broker are usually assigned. Subscribers register their interest in certain messages from one or more editors/publishers. This interest is handled through the broker, which is the responsible for managing the flow of messages between publishers and subscribers. Once the editor generates a message, it is delivered through the broker so that it sends it to the interested subscribers. Hence, the MQTT protocol is able to simplify and facilitate the synchronization between all nodes and jobs available in the IoT platform.

Each message with MQTT is associated with a topic, so the broker and the subscriber can identify the message. The usual topics are “data”, “status” or “alarm”; and they act as semantic labels of the information carried by the messages. The targeting of different topics allows administrator to check the health of the IoT solution and to monitor its network communication in a fast way.

An example of a message’s flow for MQTT is shown in Figure 4. In this particular case, a temperature sensor (publisher) is sending the temperature using the topic data to the MQTT broker. The MQTT broker delivers the topic messages to the two subscribers (computer and mobile device), which previously registered their interest in the data topic.

### 4.3. Software Architecture

#### 4.3.1. Virtualization and Orchestration

The run-time environment can be “packaged” using already known virtualization technologies [41], such as Docker [48,49]. Docker is based on the use of containers that define a prefabricated execution environment. Docker can be deployed in any infrastructure that supports this technology. The definition of a service is based on the execution of one or more Docker containers, although usually only one of them is necessary. Specifically, in the case of services associated with experimental sessions with IoT devices, the container is executed on the same device which provides sensors and runtime. However, in more advanced practices it is possible to run several containers on the same device or several at the same time. This orchestration of containers allows identifying scenarios of collaborative use where the sensors of several devices [50,51] are used in coordination to obtain a specific purpose (traffic control at crossings with several traffic lights, data from the environmental sensors of several drones flying over aerial areas for pollution indices, etc.).

Regardless of whether the service requires the execution of one or more containers, it is essential to provide an orchestration layer of those containers providing:Dynamic access management to devices. Since there may be an inherent concurrence in the development of practices in a remote environment, the orchestration layer must identify which services/containers are running on the IoT devices. This capability facilitates the search for containers/devices available in the LoT@UNED infrastructure and the assignment to new students.Redundancy and fault tolerance. The orchestration layer identifies the number of IoT devices per usage scenario (service). It is able to assign, in case of failure, a new device (or several, depending on the service). The ability to re-start (resume) the work session in case of failure is not currently supported.Management of the basic containers of the services. To facilitate the distribution of existing containers/services or the distribution of new ones, the orchestration layer should have the ability to locate the images of those containers in standard repositories [52].

There are several orchestration system solutions available, such as Docker Compose [53], Docker Swarm [54] or Kubernetes [55]. For the orchestration and control layer of LoT@UNED, Kubernetes has been selected since it eases the management of the device containers and the supervision of all the executed containers in the infrastructure through its dashboard [56]. This characteristic is essential to provide a continuous service delivery of the IoT laboratories in the infrastructure and an availability close to 24 × 7.

The main drawback of the Kubernetes deployment model is associated with the dedicated use of one of the infrastructure devices as the master node of the orchestration layer. This makes the orchestration layer vulnerable to the fall of this node and, therefore, it is essential to monitor it in real time, and to include automatic restart procedures.

#### 4.3.2. Execution Services

The complete architecture of the execution services into the LoT@UNED infrastructure is shown in Figure 5. It shows how each IoT device contains a Docker run-time environment and acts as a slave node of the Kubernetes cluster. On the control plane, there is a device (a Raspberry Pi) that acts as a master of the cluster and it communicates with the broker (IBM Watson IoT) to ease the communication channel (MQTT) with the IoT devices to be used during the interactive sessions (Shell Service).

There are currently three base containers that are identified with the “services” offered by the LoT infrastructure:*IoT*. The container provides a run-time environment based on the Python programming language and the sensor access libraries are available in device setups with the Sense Hat module. It is used in the field of knowledge of IoT solutions.*Programming*. This container/service only provides an environment with a Python distribution for its use in basic programming activities.*Security*. This environment provides the basic Linux tools for cybersecurity operations through a virtual shell console: nmap, wireshark, route, etc. commands do not really run on the provided virtual console, but directly on Raspberry Pi 3 devices, through the service orchestration platform.

With these computing services (of execution) it is possible to build and define laboratory activities in different areas using the flexible infrastructure of LoT@UNED. The fundamentals of the definition of learning services and the workload protocol to define them using the tools/applications provided by LoT@UNED are detailed in the next subsection.

#### 4.3.3. Services Implemented

In order to take advantage of the scalability of the infrastructure introduced above, it is necessary to provide such infrastructure with a set of services that allows the use of devices in remote educational environments. The offered services should implement the following features:Authenticate the user (student/teacher) into the infrastructure. It is important to facilitate the automatic identification of users. So, users must log in once but be able to access all the services transparently (SSO, Single Sign-On).Provide direct access to an interactive environment with devices. This environment is customized for the practice that the student must perform. Therefore, the actions that students can perform on the devices are limited by the environment configuration.Include analytic capabilities by storing the student’s interaction through the whole cycle with the devices. Thus, the executed commands as well as the responses can be retrieved for review. Consequently, lecturers can evaluate the student’s performance during the work session.Provide the capability to create and edit learning practices using predefined services for a specific field of knowledge (for example, IoT).

These characteristics are implemented through a set of services and applications that are included in the environment. Specifically, two fundamental applications are fully integrated with LoT@UNED:*Initial web portal for students/teachers* [57]. This website portal allows user authentication and access to the different practices available to the student, grouped by area of knowledge (see Figure 6). In addition, in the case of the teacher role, practices can be created from predefined execution services, adding the appropriate learning resources (statement of practice). Under the teacher role, work session options can also be configured (duration, commands to be executed on the IoT device, etc.). The access portal also allows verifying and analyzing the work sessions in the different practices, intending to evaluate the students.*Shell*. This application implements direct interaction with the IoT device, taking into account the possible actions and configuration of the practice defined by the teacher (Figure 7). The MQTT protocol is used for interaction with the devices, which allows the entire work session to be stored.

## 5. Experimentation

The sample case is carried out within the context of the “Cloud Computing and Network Service Management” subject, which belongs to the MSc degree in Computer Science Engineering. This degree is composed of a set of mandatory and optional subjects, some of them having 6 ECTS credits and others 4 ECTS credits. The subject considered in this work is mandatory, consists of 4 ECTS credits and is studied in the first semester of the first academic year. The degree is taught at the Computer Science Engineering School of the public Spanish University for Distance Education (*in Spanish, Universidad Nacional de Educación a Distancia*, UNED). The learning/teaching methodology is totally on distance, since Master degrees at UNED do not consist of face-to-face classes.

The subject focuses on specific competencies and skills in developing cloud computing solutions [58]. Students are provided with a guided example on the use of these technologies over a complete IoT solution. Three different and interconnected practical activities have to be solved by the students of this subject. The cybersecurity practice was not used, as it was out of the subject syllabus:Development of a simple application in a cloud service provider.Connecting IoT devices to IoT Cloud Services and Platform from a cloud provider. These IoT devices are real boards accessed remotely.Development of dashboards and data analytics based on information provided by sensors connected to IoT devices.

This solution is especially important for distance educational environments, which should satisfy the following requirements:24 × 7 availability of services associated with smart devices.Dynamic management of smart devices (horizontal growth of the IoT solution).Integration with IoT platforms and services in the cloud through the use of standard communication protocols, such as those mentioned in the previous section.Direct interaction with the sensors/elements of the devices through simple user interfaces and using Web protocols.

The following sections describe each one of the practices implemented.

### 5.1. Practice 1: Simple Application in a Cloud Service Provider

As it was mentioned earlier, the use of IoT devices is required in the second assessment. In this case, the specific skills to be learned focus on the first layer of the development of IoT solutions, this is, the sensors/devices/protocols layer. In this practice, students have to deal with three of the essential characteristics: edge programming, fog programming and protocol communication. A specific setup is integrated into the LoT@UNED infrastructure for providing students with remote access to this working layer (IoT devices). This setup is replicated, and it consists of a Raspberry Pi device and its corresponding sensors. These setups are connected to the LoT@UNED infrastructure, so they are available to the students by using the service portal. Each IoT device is able to record video and capture photos, as well as measure temperature, humidity and pressure. It also captures values associated with motion/location sensors (gyroscope, accelerometer and magnetometer), and it includes a GPS module in anticipation of future mobile scenarios.

The physical implementation of this setup is carried out with a Raspberry Pi 3, as the basis of the device/microcontroller component. This device, by default, does not have any specific sensor/actuator, but many of them can be connected to develop different projects. In this specific case, a set of additional elements has been incorporated to generate an environment with a set of sensors. These elements are:Raspberry Pi Camera. This element provides the features of video recording and photo capture. It can also be used in remote space surveillance projects, configuring the device to broadcast in real-time streaming.USB microphone. Since the Raspberry’s operating system is Debian, it is possible to connect standard devices to its ports (specifically to the USB ports). In that case, the audio recording has been added to the device to complement the video recording.GPS module. Although our learning scenario is considered to be static by default, this module has been added in anticipation of mobile scenarios. In addition to this, it provides with a very rich dataset in terms of GPS position itself, measurement error data and other associated values provided by satellites when reading GPS values.Sense Hat module. This module was originally created to work on the Astro Pi mission in the international space station. Subsequently, it became widely available to the entire Raspberry user community. The Sense Hat module (see Figure 8) provides temperature, humidity and pressure readings, as well as the values associated with motion/location sensors (gyro, accelerometer and magnetometer). Additionally, it provides an array of 8 × 8 LEDs (RGB) and a five-button joystick.

Students can access the Shell console to interact with the sensors by developing code in Python. This code is used to get sensor values (temperature, pressure, humidity, accelerometer data and GPS data). They can also write values directly on the LED array, so a word or phrase can be displayed in the array. The setup provides a video stream that can be programmed using python code (starting and stopping the video stream). To implement this activity, the related practice is designed using the corresponding runtime service “IoT”. This service is defined as one of the three runtime services available, as it was mentioned earlier.

The service is configured to connect with a Cloud IoT Service Platform (IBM Watson IoT). This way, the setup’s environment provides the MQTT library (owned by IBM and deployed on the setup) for programming and implementing the MQTT services (messages, topics and so on). These services must be deployed via Python code and consumed by an external application, which has to be developed by students (similar to the application shown in Figure 9).

Node-RED [59] is used to program this application. This framework is a visual development tool for programming IoT environments. As the setup uses MQTT, students have to use connectivity blocks for sending and receiving MQTT messages or topics. On the one hand, this tool allows the subscription to one or several topics of the MQTT message system (coming from one device, or several ones); such as the data from the sensors, current sessions or stops and messages between them. On the other hand, a topic (message or event) to the MQTT message system for session management can be published.

### 5.2. Practice 2: Connecting IoT Devices to IoT Cloud Services and Platform from a Cloud Provider

After developing the “local” solution, corresponding to the sensors/devices/protocols layer, students must learn and know the operation and services of an IoT platform in the cloud. In this case, it is specifically intended that they learn how to store data from the sensors they are using in the local solution. Since MQTT is used as a communication protocol, any cloud service platform that supports this protocol can be used in this part of the learning scenario (layer 2 of the IoT’s full development model). The platform used by the students for their practices is IBM Watson IoT because the LoT@UNED infrastructure itself is based on this platform.

The main objective of this activity is to become familiar with the use of a series of services offered by IBM Watson IoT, focusing on the storing of sensor data and device management. IBM Watson IoT has a management space for device types and registered devices. Again, to understand the services provided by the Cloud IoT Platform, a student must use a specific activity defined in LoT@UNED. This practice is based in the “IoT” runtime service and its goal is to connect with the management space (using MQTT protocol) and check the services for this Cloud IoT Platform. The full documentation and services description is available in [60].

Additionally, to provide a cloud storage service, students must develop a single cloud application, which uses the Cloudant [61] service to store the sensor ’s data. This application uses Node-red framework to facilitate the integration with the MQTT protocol and get the data from the device (assigned using LoT@UNED infrastructure). The Node-red distribution, included as a service in the IBM Cloud platform, has specific blocks to connect with Cloudant services to simplify the storing of information (see Figure 10). This data will be used in the next step of the learning scenario for the Layer 3 of our sample case.

In short, the practice focuses on the aspects related to the specific communication protocols of IoT and the integration with external suppliers. In addition, students experiment with the security mechanisms of these protocols and the applications/services that use them. For the essential characteristics, students work on: protocol experimentation and cybersecurity.

### 5.3. Practice 3: Development of Dashboards and Data Analytics

In previous sections, the theoretical aspects of dashboards and assisted decision layer learning were given and, also, how this layer is linked with the LoT@UNED platform. Now, we detail a concrete example of an application for this layer.

The presentation and decision layer provides human-readable information to see what is the status of the IoT solution (specifically, status and information data from sensors). Sensors produce valuable information from the environment in which they are integrated. This information allows the generation of indicators to monitor different types of environments where sensorization can be critical. For example, in the case of medical environments, biomedical sensors allow information to be collected and displayed on dashboards to monitor patients [62,63]. The importance of the development of these dashboards depends on the information monitored, but usually, at least, a dashboard is developed to have monitoring information of the IoT environment. As previously seen, the information from the environment is stored in a data storage service that is usually in the cloud. This information can be represented in real-time, by dashboards, or analyzed to calculate performance indicators. These indicators can be used in decision-making and risks evaluation [46]. Therefore, these decisions are assisted by IoT data.

In this particular practice, the student will work on the development of a dashboard that uses the analytical capacity of the cloud providers and will represent the relevant information from the IoT data. This way, students will work on the essential characteristics corresponding to cloud dashboard and analytics programming. The dashboard must show real-time information about temperature, humidity and pressure (provided by the Sense Hat sensors). In addition, other indicators can be shown dynamically, such as time, the accelerometer values (X, Y and Z coordinates), pitch, yaw and roll. Figure 11 shows an example of a single dashboard built with basic gauges. This example is a basic template provided to students, which has to be modified and enriched following basic visualization techniques.

Once the student has proven to be able to solve IoT data representation problems, they have to add the decision element in the production line related to an IoT solution. These decisions must be based on using stored data from sensors, and they will vary depending on the environment in which the solution is integrated [64]. The use of instantaneous data is not enough, so it is necessary to verify the evolution of the data and calculate performance indicators (usually statistical indicators). The cloud IoT service platform stores the data (layer 2 in our model), so the indicators can be analyzed and graphically represented. Data being in the cloud allows for its use in different service providers that have data analysis tools and in many cases, given the nature of the IoT data production ratio, Big Data techniques must be applied.

On the other hand, by using the Apache Spark analytics service, students have to analyze the stored data. Another service named IBM Watson Studio is used jointly with the Apache Spark Engine to facilitate the use of the analytics service. These services allow for the creation of notebooks in a variety of programming languages (among others, Python or Scala) for interactive work with the aforementioned services. Students must use one of the following sensed parameters as a basis of the analysis: temperature, humidity, or pressure. Additionally, some filter tasks are necessary for data. Some examples are changing the sensed time from string to DateTime, grouping values, filling empty values, transforming data to make specific accesses or truncating values. This way, students learn how to manage the generated data in the cloud and prepare them for the data analysis itself.

## 6. Results

This section analyzes the data obtained from an opinion survey provided to LoT@UNED students. Some preliminary results and conclusions were included in [65]. The amount of surveyed students was 129, in which 89.15 % of the users were male and the 10.85 % of them were female, as indicated in Table 2. With regard to the job occupation, a big amount of students are not related to computer science. In particular, a total of 79.9 % of students.

Figure 12 shows the comparison of the job situation of the surveyed students about the LoT@UNED platform, in terms of their job profile (computer scientist, non-computer scientist and others) with their age divided by ranges (less or equal to 30 years, between 30 and 39 years, between 40 and 49 years and equal to or older than 50 years). As observed, many students are in the range of 30 and 39 years old with a dominant computer science profile. The conclusion about the job occupation is even stronger for the range of 40–49 years old. In contrast, the youngest and oldest students have an occupation profile out of computer science.

The measured indicators were the perceived usefulness of the LoT@UNED platform by students, its ease of use for practical activities, the users’ attitude when using the platform, the social influence when using it, the ease of access to the platform and the students’ intention of use the platform for practical activities within the context of LoT@UNED.

Table 3 represents the statistical data generated from the students’ opinion survey (perceived usefulness, ease of use, user attitude, social influence, ease of access and intention of use), in terms of the standardized mean, standard deviation, variance, minimum and maximum values, median, kurtosis, asymmetry and Cronbach’s alpha. Regarding the mean values of indicators, with a five-point scale, they can be considered as very good. The best one is the ease of use with a value of 4.13, but the worse one is the ease of access with a value of 3.40. This fact can be due to the student’s profile described above. The presented standard deviation and variance values are not so high, enforcing the goodness of the exposes results. In addition to this, mean and median values are very similar. The analysis of the kurtosis, asymmetry and Cronbach’s alpha indicators indicate that these results are consistent.

The kurtosis characteristic describes the concentration of data around the average of each indicator shown in Table 3. These kurtosis values are positive for four indicators (they are on the right side of the mean) and negative for two of them (they are on the left side of the mean). These characteristics consider the standardized mean of each indicator as a central point, so the data distribution is close to each indicator mean. This means they are not too scattered and in ranges of normality. This is enforced by examining the median value of each indicator, since they are near its corresponding mean.

On the other hand, the asymmetry characteristic measures the degree of symmetry of the data distribution for each indicator shown in the horizontal axis. These asymmetry values are negative in all cases, except one of them, so their distribution generally tends to the left within the x-coordinate axis. Obviously, the positive case is to the right side. They are not too high of values, so they are considered as a good distribution.

In addition to this, the Cronbach’s alpha for each indicator is bounded among 0.87 and 0.90. These values are considered as more than acceptable. What is more, the general Cronbach’s alpha is slightly higher than 0.9. This means that the reliability of all indicators together is really good, and we can conclude that there is a correct internal consistence. The Cronbach’s alpha calculates the mean of the correlation among the exposed indicator.

These results are very rich, since they contain lower and higher values, as indicated with the minimum and maximum values. To sum up, the exposed statistical values are satisfactory, by considering the literature [66,67], being very reliable to be employed in further studies. Additionally, Table 4 shows the amount of students who answered for each indicator: strongly agree, agree, neutral, disagree or strongly disagree.

Finally, Table 5 indicates how the selected indicators are correlated among them. The represented values enforce the conclusions obtained for the statistical data described above. There is a strong influence among them. The next step would be to examine their concrete influence, and how they are related. The perceived usefulness influences the user attitude about using the LoT@UNED platform in very a strong way, with a value of 0.813. The usefulness indicator also affects the intention of use of this platform in the future in an indirect way. This value is 0.689. Another strong influence is the user attitude versus the intention of use, with a value of 0.768. The rest of the indicators are very influenced among them in a lower manner. Results marked with * correspond to the presented ones in the same table when comparing the two indicators in the opposite axis.

## 7. Discussion

According to the results obtained in the student survey, students find the LoT@UNED platform very useful (standardized mean for perceived usefulness: 3.93). It is a global indicator associated with the activities carried out in the LoT@UNED platform, described in the experimentation section. Being a non broken down indicator makes it impossible to get a particular statistical measure for every essential characteristic. However, taking into account the population associated with the survey and the statistical result, we can infer that the platform is useful for implementing each of these essential characteristics (edge, fog, cloud and analytics, protocol and cybersecurity). This inference is based on the fact that the design made for the three practices implements the five essential characteristics.

It is also interesting to note that the ease of use and intention of use have high values (standardized mean: 4.13 and 4.04, respectively). This issue means that the design of the platform itself has been done correctly and has simplified the development of the practices carried out by the students. Moreover, the high value of the intention of use indicator allows inferring that the student would be willing to use it in more similar practices in the IoT laboratory environment and even in other disciplines (cybersecurity, programming, etc.). This ability is possible because the platform allows the generation of specific services supported by the set of IoT devices that compose it.

The lowest values of the indicators (even though, they are values that indicate good behavior) correspond to ease of access and social influence indicators (standardized mean: 3.40 and 3.67, respectively). From these values, it can be deduced that the laboratory’s workflow and the way to access the devices may be improved. This issue mainly affects the essential features of edge and fog computing because the virtualization layers introduce delays and complexity in the interaction that influence the ease of access and interaction. Additionally, the social influence indicator warns about the lack of interaction between students inside and outside the LoT@UNED environment. Practices are indeed carried out at distance and individually, so the social factor has less influence than in a face-to-face environment, but it is necessary to work on it. For this reason, the platform must provide collaborative tools that facilitate social interaction and communication in real time between students and teachers. These new features will be included in future versions of the platform. These improvements will focus more on the design part of learning than on the development of IoT lab environments and the support of essential characteristics for educational IoT laboratories.

## 8. Conclusions

The learning/teaching processes in the development cycle of IoT solutions imply a set of skills ranging from devices and IoT sensors, their communication protocols, the storage management and the processing environments on the Cloud for data generated by sensors. These environments are then eventually able to make decisions or show the relevant information on those sensors (as indicators). These fundamental competences are needed in the full cycle of development of IoT solutions, consisting on three layers: (1) basic interaction with sensors and specific communication protocols; (2) data management models to handle the generated data; and (3) processing and visualization of the most relevant indicators on these IoT devices. In this last step, the processing can include a specific communication protocol. This protocol could be used to perform actions in the IoT device itself as a response to the processed indicators (for example, using available actuators at the device).

According to this, this work first presents the main features of the LoT@UNED platform, which has been developed to cover the instructional design of our subjects, and how the three layers of the proposed full cycle of development for IoT solutions are implemented in it. The essential characteristics for this kind of laboratories/environments are fulfilled by this platform: edge programming, fog programming, cloud dashboard and analytics programming, protocol experimentation and cybersecurity. Each phase is associated with a specific activity that is deployed in a standard way using Docker containers managed through a cluster manager (with Kubernetes). The manager balances the workload of different devices. Thus, the use of the devices/sensors is assigned in a dynamic way to the students who are developing the activities. This platform allows students to implement all these phases efficiently and redundantly, providing high availability for its use.

The proposed LoT@UNED platform has also been used for students in several computer science subjects. The use of this platform is especially relevant in online educational environments, as is the case of distance universities. This way, they perform remote experimental activities with a collaborative IoT learning infrastructure in the cloud, analyze the data generated and make visual representations in it. As for the result and discussion sections, we can conclude that the perceived usefulness and ease of use of the proposed platform values are really good, as well as the intention of use it in the future for additional practices. The students’ attitude is also great with respect to the use of the platform in practical activities. The rest of the indicators are good, although they are challenging for working on improving the social influence among students when using it, and easing the access mechanisms.

As for future work, the presented method for validation of the IoT platform will be improved. To achieve this, a UTAUT model will be hypothesized. The same set of factors will be considered (easy of use, usefulness, attitude, social influence, …) to be included in this model, in order to check the intention to use the presented technology. Another future line of research is to exhaustively analyze the students’ learning progress into the LoT@UNED platform. Finally, the source code of this tool has not yet been shared with any other institution but the release of the code is also one of our next steps for future work. We would like to have a community around it to go on including improvements.

## Figures and Tables

**Figure 1 sensors-20-03770-f001:**
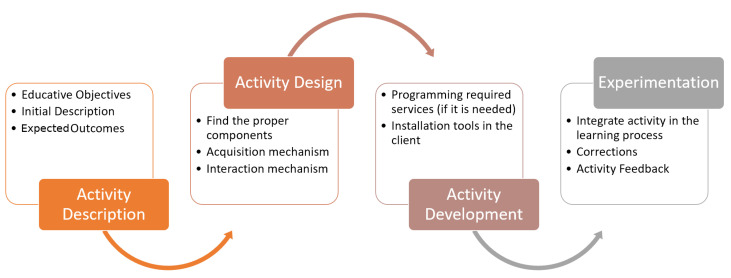
Instructional design phases for a new laboratory.

**Figure 2 sensors-20-03770-f002:**
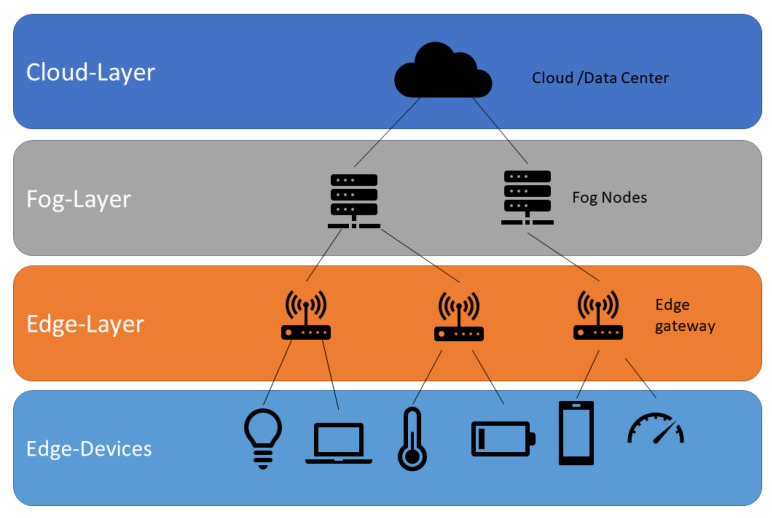
Fog computing approximation for Internet of Things (IoT) solutions. Figure available on [34].

**Figure 3 sensors-20-03770-f003:**
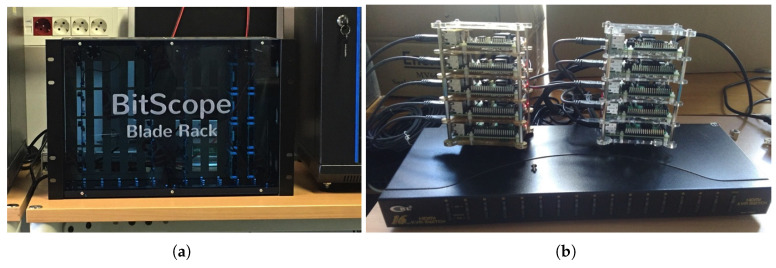
Devices clustering: blade rack and cheap setup. (**a**) Blade rack. (**b**) Cheap setup.

**Figure 4 sensors-20-03770-f004:**
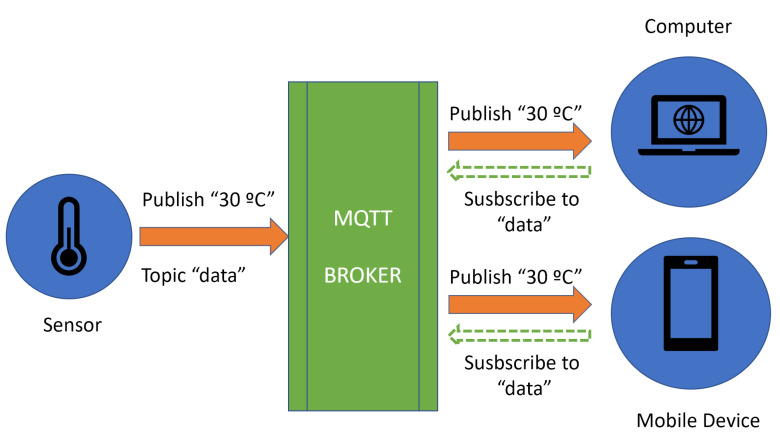
MQTT flow of messages for topics and subscriptions.

**Figure 5 sensors-20-03770-f005:**
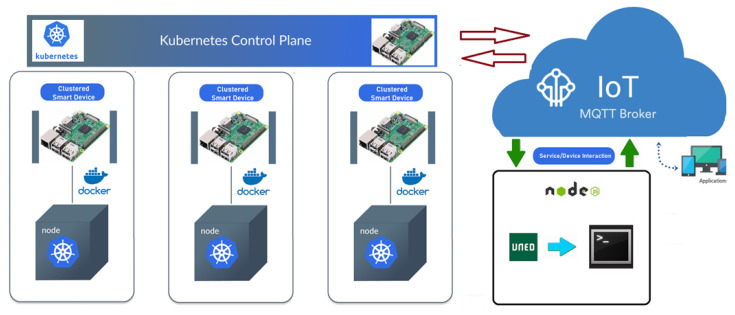
Technical solution for Labs of Things at UNED (LoT@UNED).

**Figure 6 sensors-20-03770-f006:**
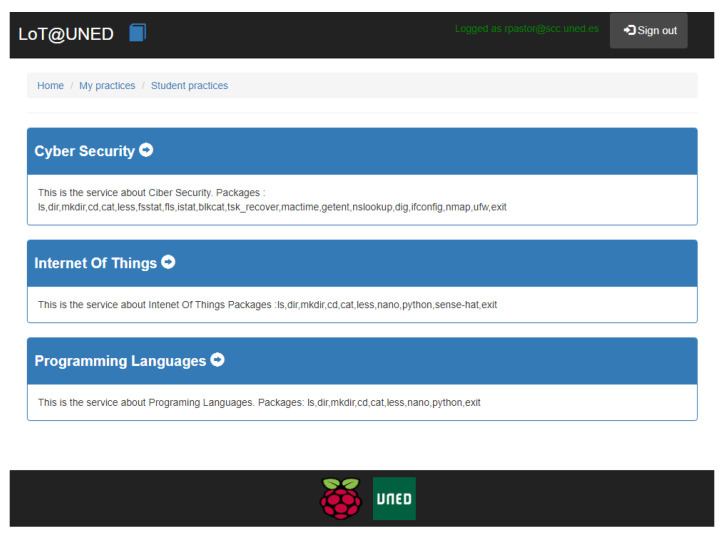
Knowledge domain for IoT supported learning scenarios.

**Figure 7 sensors-20-03770-f007:**
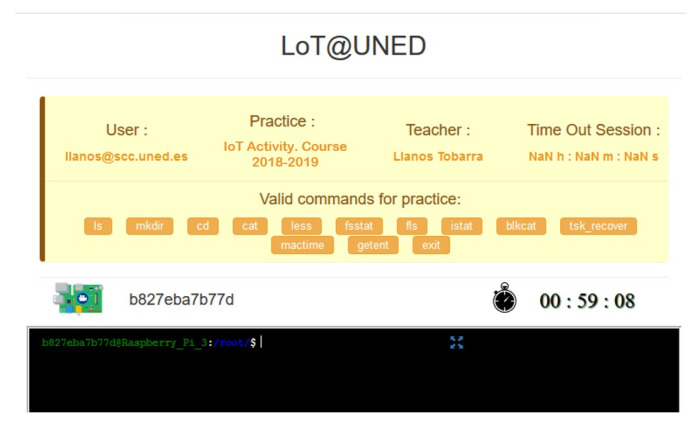
Interaction with IoT devices using the Shell service in LoT@UNED.

**Figure 8 sensors-20-03770-f008:**
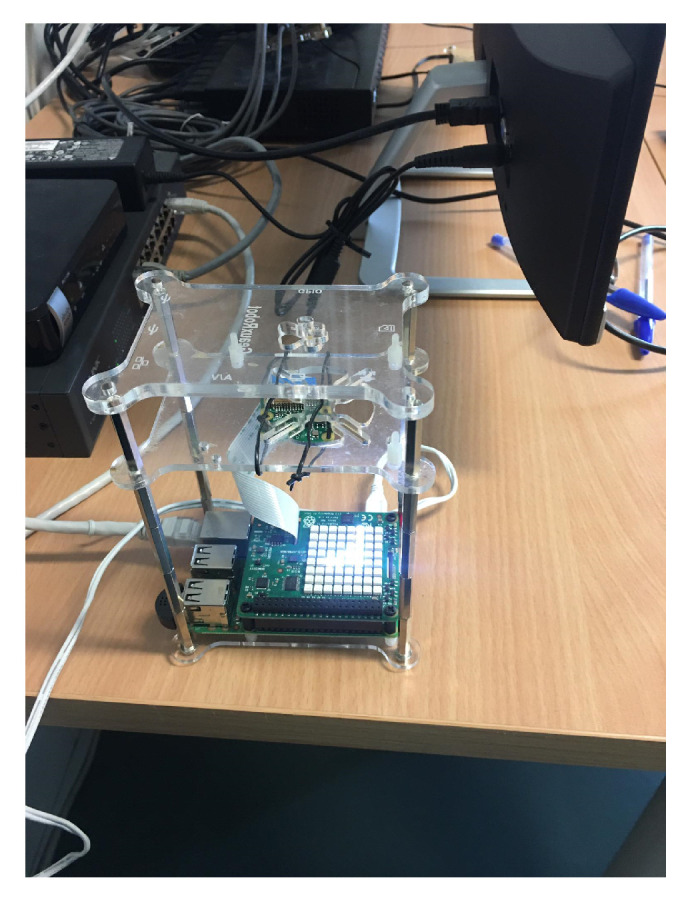
Example of individual Raspberry Pi device.

**Figure 9 sensors-20-03770-f009:**
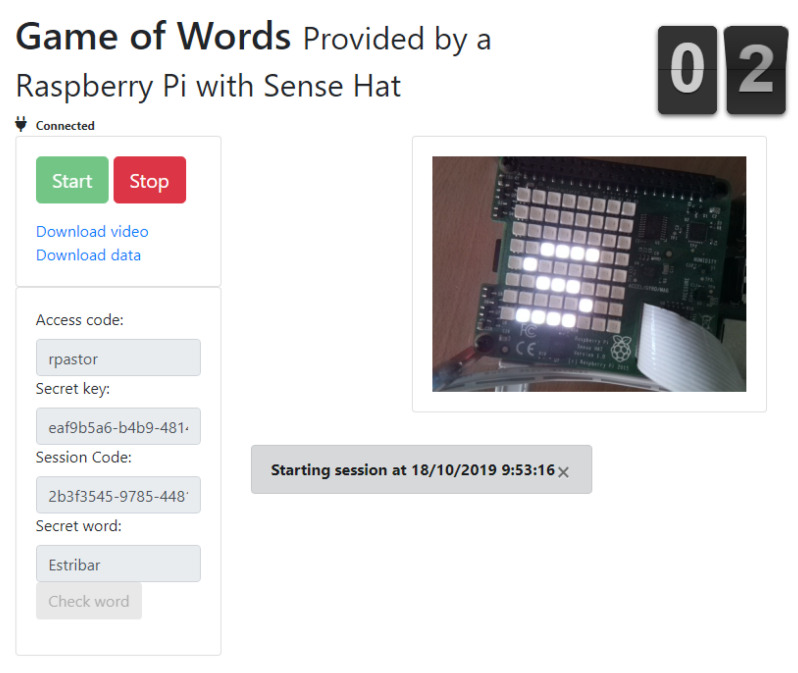
Game of words. IoT devices with sense hat.

**Figure 10 sensors-20-03770-f010:**
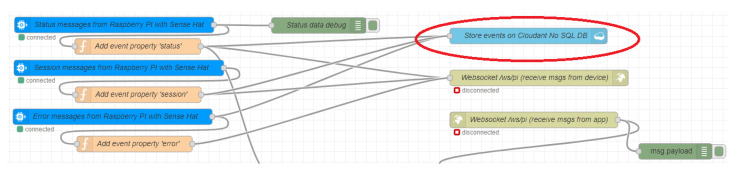
Cloudant integration in a Node-Red application.

**Figure 11 sensors-20-03770-f011:**
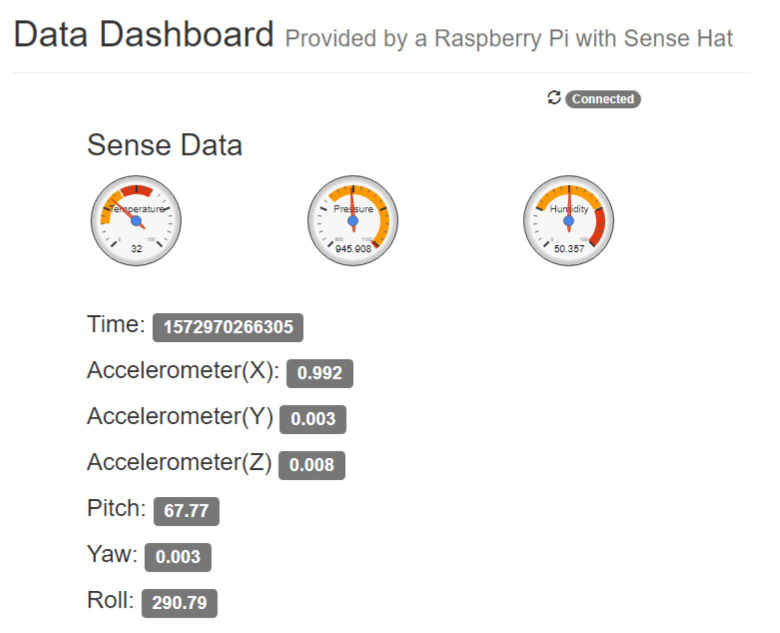
Single dashboard for sensor data.

**Figure 12 sensors-20-03770-f012:**
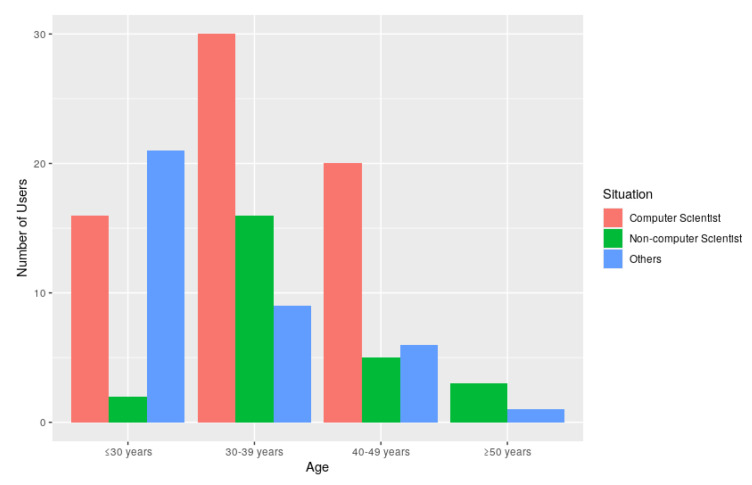
Comparing the job situation versus the age ranges for users who tested the LoT@UNED platform.

**Table 1 sensors-20-03770-t001:** Functionality comparison of the state of the art on IoT remote labs.

	Edge Programming	Fog Programming	Cloud Dashboard and Analytics Programming	Protocol Experimentation	Cybersecurity
Tunc [41]					X
El-Hasan [42]	X				
Fernandez-Pacheco [43]	X				
Leisenberg [44]		X			
Rajurikar [45]	X		X	X	
Authors	X	X	X	X	X

**Table 2 sensors-20-03770-t002:** Users’ profile.

Indicator	Options	(%)
Gender	Male	89.15
	Female	10.85
Age Group	≤30 years	30.23
	30–39 years	42.64
	40–49 years	24.03
	≥50 years	3.10
Occupation	Computer science related job position	20.1
	Non-computer science related job position	51.2
	Others	28.7

**Table 3 sensors-20-03770-t003:** Results obtained from an opinion survey after testing the LoT@UNED platform (statistical data).

	Perceived Usefulness	Ease of Use	User Attitude	Social Influence	Ease of Access	Intention of Use
*Standardized Mean*	3.93	4.13	4.11	3.67	3.40	4.04
*Standard Deviation*	0.87	0.95	0.89	0.79	0.71	1.04
*Variance*	0.76	0.91	0.80	0.63	0.50	1.09
*Minimum Value*	1.00	1.00	1.00	2.00	1.50	1.00
*Maximum Value*	5.00	5.00	5.00	5.00	5.00	5.00
*Median*	4.00	4.33	4.25	3.67	3.50	4.33
*Kurtosis*	0.96	0.15	0.43	−0.92	−0.04	0.01
*Asymmetry*	−0.95	−0.96	−0.93	0.21	−0.27	−0.96
*Cronbach’s Alpha*	0.88	0.89	0.87	0.89	0.90	0.88

**Table 4 sensors-20-03770-t004:** Results obtained from an opinion survey after testing the LoT@UNED platform (counting with a five-point Liker-scale).

	Strongly Agree	Agree	Neutral	Disagree	Strongly Disagree
*Perceived Usefulness*	43	58	20	7	1
*Ease of Use*	61	37	21	9	1
*User Attitude*	61	40	21	6	1
*Social Influence*	22	46	55	6	0
*Ease of Access*	7	63	50	9	0
*Intention of Use*	61	34	19	13	2

**Table 5 sensors-20-03770-t005:** Correlation matrix among the exposed indicators.

	Perceived Usefulness	Ease of Use	User Attitude	Social Influence	Ease of Access	Intention of Use
**Perceived Usefulness**	1.000	0.597	0.813	0.615	0.552	0.689
**Ease of Use**	*	1.000	0.609	0.527	0.582	0.597
**User Attitude**	*	*	1.000	0.616	0.610	0.768
**Social Influence**	*	*	*	1.000	0.517	0.543
**Ease of Access**	*	*	*	*	1.000	0.535
**Intention of Use**	*	*	*	*	*	1.000

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
