# Peer review of "A WoT Platform for Supporting Full-Cycle IoT Solutions from Edge to Cloud Infrastructures: A Practical Case"

_sensors, 2020, doi:10.3390/s20133770_

Round 1
Reviewer 1 Report
The authors present a study motivating that the increasing importance of IoT technologies demands for more professionals with adequate knowledge to deal with these heterogonous and complex environments hence the need for proper education environments and course that can support the specifies of this multidisciplinary area of knowledge. The authors present an overview of the areas that need be taught as part of IoT ecosystem, they introduce the LoT@UNED learning environment designed for this purpose and they exemplify this with a case study on a specific subject.
It is the second time that I review this paper and I think it has greatly improved from previous iteration. I also think that the special issue on “Teaching and Learning Advances on Sensors for IoT” is a great match for the contribution. This work can provide useful ideas to educators and universities that teach about Sensors and IoT.
I have some additional comments:
It would be good to discuss about the pedagogical implications and learning outcomes of using this platform in the classroom. What do students and teachers think? Perhaps it can suffice with an additional paragraph in the conclusions.
Is this platform open somehow? Or could other universities use it or install it as a local instance?
Use center alignment for Table 1, right now the numbers are in a corner and that provokes confusion.
Typos:
“Stare of the art” State
“an remote” a remote
“in spanish,” Spanish
Author Response
Thanks for your comments. Please, see the attached document.
Kind Regards

Reviewer 2 Report
The paper describes a very interesting platform, which can be applied for teaching. There is a very good review of the state of the art, providing a good number of references.
The stated objectives cannot be to present the state of the art and the platform. Authors should reflect on what the proposed platform advances the state of the art and formulate the objectives accordingly. In addition, the presentation of the state of the art and the relationship with the proposal should be a part of almost all papers, but this should not be an objective, unless this could be a review paper, which is not the case.
Methods should not only be of the state of the art but related to the whole research.
The paper is easy to follow but needs a review of the English language.
Author Response

(The authors gave the same response as above.)

Reviewer 3 Report
This work proposes a platform for student education, which they named Web of Things (WoT) platform named LoT@UNED. Then they introduce how it works.
- After reading, it is more like to report rather than research articles. First, the research question is unclear, and most of the work is to introduce the platform with some cases.
- The state-of-the-art needs to analyze how it related to the platform.
- The evaluation part is not convincing, as there are real results presented. Most are to introduce how to setup the platform.
Author Response

(The authors gave the same response as above.)

Round 2
Reviewer 3 Report
No more comments.
This manuscript is a resubmission of an earlier submission. The following is a list of the peer review reports and author responses from that submission.
Round 1
Reviewer 1 Report
The authors present a study motivating that the increasing importance of IoT technologies demands for more professionals with adequate knowledge to deal with these heterogonous and complex environments hence the need for proper education environments and course that can support the specifies of this multidisciplinary area of knowledge. The authors present an overview of the areas that need be taught as part of IoT ecosystem, they introduce the LoT@UNED learning environment designed for this purpose and they exemplify this with a case study on a specific subject.
I think the article is a good match for this specific section of Sensors journal, the topic is important and the authors make a good job describing a good number of ideas that can be useful for educators or institutions that are teaching or will be teaching IoT technologies.
Suggestions for improvements:
In the abstract “These are the three main steps to follow.” And what follows afterwards, is presented as an absolute truth. I would use a language that clarifies that this just your proposal and there might be other approaches. I would clarify this in other sections of the manuscript as well.
Can you extend on what which learning pedagogies can be appropriate for this specific scenario of IoT technologies and the decisions with respect these pedagogies in the design of the course?
Grammar:
In general, the manuscript should be slowly reviewed as there are numerous typos, some examples:
“Fog computing [4,5] that add” – adds
“which not feasible” – needs verb
“Lot@UNED” missing capital T
Author Response
We like to thank you for your constructive comments. We think the new paper version is better thanks to you. Please, see the attached word document with our replies to your comments.
Best Regards

Reviewer 2 Report
1. The contribution of this paper is not sufficient, and it is not SCI level paper but more like a undergraduate graduation thesis. 2. The theoretical analysis and formal relevant model(s) is not provided. 3. The algorithm and simulation part is missing. 4. Overall, the paper is low level and does not match SCIE journal level.
Author Response
We like to thank you for your comments. We think the new paper version is better thanks to you. Please, see the attached word document with our replies to your comments.
Best Regards

Reviewer 3 Report
The title in its present version prepares the reader for reading about learning the development of full cycle solutions in IoT environments; however the paper in its core, presents – along with processes related with learning- the features and characteristics of LoT@UNED platform. Under this scope, I would propose a revision of the title or amendments in the core content of the paper so as to map in a more successful manner the work presented.
The Abstract could map in a better manner the content presented at the core of the paper; after the first sentence on IoT , the authors make clear reference to (3) specific phases/steps which however, do not map exactly with the rest of the paper’s content. A suggestion would be authors – since they focus on the use of LoT @UNED platform – to briefly describe the features of this platform, and connect it with the learning design methodology they have chosen and applied – which could also be robustly presented by justifying it to specific use of literature ( education and IoT solutions’ value) in later section of the paper. At the Introduction (not necessarily only in Introduction) authors could use a more diplomatic language ( ie “To make these changes… must be implemented”) by rephrasing the content.
In the specific section, the authors have included a paragraph about Fog computing- however ,this could be inserted in later sections of the paper since in its present version the use of it is not clear to the reader, neither the purpose of its use. In the fourth paragraph, the authors refer in a general manner on the use of IoT applications in everyday life; however, if this paper’s focus is on learning and educational domain, it would be a good idea to include a respective paragraph with teaching and learning approaches in IoT , justified by relevant literature, and justifying their choice also on the specific research design and methodology. A suggestion would also be to move the Fog computing section along with the paragraphs 5 and 6 of the Introduction in later sections of the paper, so as to further support their research design and methodology in a more robust manner.
The Layers and Components of IoT section could be condensed ; the protocols could be presented but there has to be a focus on the authors’ choices regarding the applied methodology and research design which is not clear by the present form of this section. Some grammar inconsistencies have occurred at this section ( ie “ The subscriber registers their interest…”). If included, the third paragraph of this section could be schematized; the way content is presented the relevance of it with the focus of this paper is not clear- content has to be filtered so as to “map” to the specific work presented, avoiding generally presented information, which are important, however at times they disorientate the reader. In the 3rd section ( WoT platform for learning in developing IoT solutions : LoT @UNED) it is important authors to a) highlight the value of the specific platform, b) refer to similar efforts already made in this field and how their work is differentiated or is similar to these. A more segmented version in subsections of the sections included could enhance the readability of the paper and make the content more easy for the reader to follow and grasp ( ie platform’s design rationale, platform’s infrastructure). If the “learning” focus is kept in the Title the 3.3 Learning Services section needs to be more extended, providing the core of this paper. In its present form in this section some grammar inconsistencies have occurred ( ie “that can be take place in a Laboratory”). The authors in this section refer to the use of “ a general learning design methodology must be applied” ; however, this is one sentence and authors need to justify it with respective literature and extended content towards this direction, what do the authors mean by “ a general learning design methodology”, what is that, how is it justified in literature and how they have implemented this in practice? A suggestion would also be authors to clearly discriminate sections of the platform infrastructure and what the tutors/ students can do with the platform- this would enhance the readability of the paper. The title of section 4.1 could change to “Educational context”.
The use of past tenses is quite important in describing work, which has already been implemented. The phrase “ The subject has three assessments” needs rephrasing. In the 4.2 section, in the first sentence it is not clear what is “the second activity”- authors could revise their paper so as to present the content in a more cohesive manner. The inclusion of more schemata in section 4.4 could also help. My suggestion is a) authors to try and group the learning processes undertaken with the use of specific platform, b) present how the specific platform supports these processes, in a focused manner.
Author Response

(The authors gave the same response as above.)

Round 2
Reviewer 2 Report
1. Please use full name of UNED with country name. 2. This paper is poorly written and there are so many grammatical mistakes even in the 2nd round version. For example, in the Abstract: "...authors proposes..." -->"...authors propose..."; ("Phase 1 corresponds with... phase 3 must be focused on..." very poor expression!) 3. The main contribution is not clear and enough. 4. Section 2 and most part of Section 3 are tutorial knowldge. 5. There is no theoretical analysis and simulation part is very poor (without comparison). 6. Ref. papers 1-31 are very briefly explained, which shows the authors are very non-professional. In conclusion: this paper should be rejected withtout further processing in Sensors.
Author Response
Dear reviewer, thanks for your comments. Please, you can see the attached document with our responses.
Best regards

Reviewer 3 Report
The title of the paper has been revised and in its present form conveys in a clearer manner the content presented in the rest of the paper. Overall the structure of the paper is much clearer, conveying in a more concrete manner the respective content of the paper in relation with the former version of the paper. However the paper needs further attendance , in my view, in language issued and content presentation, at specific points. In the Abstract section there are some points that could be better expressed ( ie “ propose a learning design cycle in several phases that allows students …” “the proposed cycle consists of the following steps”. In line 7 of Abstract “the” use is not justified by the language context provided- description of Phase 2 in Abstract is not clear. Better use “ performed” than “ done by algorithms” . Language problems again in lines 35-37 could be resolved by rephrasing and presenting in a clearer manner respective content.. Again in line 78- 80 language problems do exist and the content presented is not clear- it needs rephrasing. What about section 1 description of the paper? Its description has not been presented in line 74. In section 2.1 at specific points the respective content is not clearly presented . The sentence in lines 109-112 is hard for the reader to follow and needs rephrasing. The same applies for the sentence presented from line 114-118. The connection of presented protocols with M2M could be better presented in section 2.2.1 . The sentence between lines 321-323 needs rephrasing since its content is not clear to the reader. The title of section 3.4 I would recommend to be supported by a word following the “learning design “synapsis: ie learning design cycle etc. The sentence presented in lines 422-424 is hard for the reader to follow and needs rephrasing. The same applies for sentence in lines 455-457. What do the writers mean by “ the subject was composed by three assessments”? The meaning of this sentence is not clear. In line 507 the adjective’s number does not coincide with the number of the noun. The methodology section of learning design could include a more elaborated and justified presentation of important pillars that led authors to specific decisions. Section 4.4.1 needs a better connection with the earlier parts of the paper and especially the last one. The same applies also for section 4.4.2 – these sections do not seem organically related to the earlier parts of the paper. I would recommend authors to use as title of the last section “Conclusion and Future Work”. This section it is my suggestion to be revised, reflecting the content of the whole paper, in a recap manner.
Author Response
Thanks for your comments. We think the comments make a better paper and we like to thank you for being so constructive.
Please, see the attached pdf document with our response.
Best regards

Round 3
Reviewer 3 Report
In the Abstract section the authors refer to “proposed cycle”- however from the language used in the respective section, it is not clear to the reader, what “the proposed cycle” refers upon and how it is connected with IoT learning; if the focus of the paper is the platform presentation, there has to be a respective focus in the Abstract section on this, by elaborating a little bit more on the use of the platform. In the Abstract also authors refer to “learning design”- however this is a broad term and it is not evident to the reader what exactly do they mean by learning design and how the use of the proposed platform encapsulates aspects of the learning design. So, under this scope, the Abstract needs further revisions in order to achieve the best of readability and content clarity. While in the title authors refer to “ WoT platform” in the last line of the Abstract they refer on “ an IoT learning environment”; it is my view that there has to be a consistency in the terms used, applied through all the paper. In the Introduction section the second paragraph lacks connection with previously presented content; a major revision is needed in the transition from paragraph one to paragraph two as well as in the content of the second paragraph since information presented , seemed a little bit scattered. The Introduction section needs major revision in the sense of a) presenting in a brief manner the respective theoretical framework of work presented, b) setting some important methodological issues that this work makes use of, c) present in a clear manner the focus of this work. In the section “Layers and components of IoT solutions” authors have to provide some kind of theoretical information on how this section relates with the already presented and content but also the next sections of the paper- further attendance in cohesion of content is requested. The use of literature resources could be more extended at the specific section, justifying claims and terms used ( ie “scenarios”). Language used needs further attendance (ie “the ability to add advanced processing or storage features”- to what?). What is the relation of “communication latencies” with the focus of this work? Examples of “communication latencies” could also be included. The content adhering to Fog and cloud computing has not been clearly presented in terms of their exact use for the specific platform- authors could provide a more processed overview of Edge, Fog and Cloud focusing on commonalities and differences but also showing how these relate with their applied approach in developing the platform. This means that the methodological underpinning of the platform presented has to be more robustly presented in terms of the already provided comments. Authors present three layers in IoT solutions; however from the presented content, it is not clear how do these relate with the described platform, how do these map with the respective platform- so elaboration towards this direction is needed. The definition of M2M as a term could be more precise and making use of existing literature as well as elaborating on which protocol has been used for the specific platform development. It is my view that the section “ Cloud IoT services platforms” could be presented in earlier sections of the paper as a preview of the description of services provided by the specific platform. It is not evident to the reader how the information presented in the “Dashboard and Assisted Decision” section exactly relates with the specific platform’s development; it would be a good idea authors to filter the information and theoretical content provided in the paper, according to the specific platform’s characteristics. In earlier sections authors have referred to “learning design” and “scenarios”- however up to section 3, there is no elaboration on respective methodology used or how learning schemata have been used for the purpose of this work. All the information provided so far have been technically oriented (description of IoT infrastructure) with no reference to the pedagogical/educational aspect . How does the use of “grouping” relate with the “learning design”? From section 3 authors present the platform, however no connection has been made with already presented content ( in earlier sections of the paper)- content has to be further processed in terms of these parameters. It would be a good idea authors to include references when referring to “learning design cycle” (section 3.4) and provide more information regarding the exact methodology used. In the 4.2 section, authors mention that “ the use of IoT devices is required in the second assessment”- however this is not clear to the reader. The content of section 4.2 is important to be filtered according to already provided information, in earlier sections of the paper. Authors it is also important to justify their claims by the use of specific literature ( ie 4.4 “The presentation and decision layer is important”- justification from literature is needed at design decisions and claims. Section 4.4. is not very clear to the reader ( what has been the reason for describing that) so it is important to filter its content according to already presented information. Language used needs further processing at almost all the sections of the paper so as a) the scope of the paper to be clear, b) the content of the paper to be coherent. Diagrams should be presented in earlier sections of the paper and not in the Conclusion section. The conclusion section needs to be more elaborated in the sense of revising important information presented in the paper. Overall, it would be my recommendation a) authors to support their methodological claims (as presented in the comments provided), b) filter the content of the paper according to the scope/focus of the specific paper, avoiding too much theoretical information in earlier sections of the paper, c) use appropriate language with condensed but well processed sentences so as to enhance readability of the paper.
Author Response
Dear reviewer, thanks for your valuable comments and suggestions. You can check our changes to the paper and comments in the attached pdf.
Best regards
